# Algorithm for Application of a Basic Model for the Data Envelopment Analysis Method in Technical Systems

**Mariia Pokushko** [1,2,3,*], **Alena Stupina** [1,3,4], **Inmaculada Medina-Bulo** [2], **Svetlana Ezhemanskaya** [1], **Roman Kuzmich** [1] **and Roman Pokushko** [3]

1   Institute of Business Process Management, Siberian Federal University, Krasnoyarsk 660041, Russia
2   Superior Engineering School, University of Cadiz, Puerto Real, 11519 Cadiz, Spain
3   Institute of Informatics and Telecommunications, Reshetnev Siberian State University of Science and Technology, Krasnoyarsk 660037, Russia
4   Siberian Fire and Rescue Academy of GPS MES, Zheleznogorsk, Severnaya 662972, Russia
*   Correspondence: mari.pokushko@alum.uca.es

**Abstract:** The aim of this study is to solve the problem of increasing the efficiency of fuel and energy complex enterprises. Because such enterprises are complex systems, it is difficult to optimize their work, taking into account all the technical indicators of such enterprises. This study proposes to solve this problem by defining an algorithm using the data envelopment analysis (DEA) method. In particular, the algorithm was applied in heating systems using the example of a combined heat and power plant, where the DEA method had not previously been used. Experiments were carried out to improve the efficiency of the combined heat and power plant. Efficiency indicators were calculated, changing inputs and outputs of the model according to the study case to achieve the maximum efficiency of the system. The Charnes; Cooper and Rhodes; and the Banker, Charnes, and Cooper models were tested with good results. The presented methodology and experimental results enabled the DEA method to be applied for the first time to improve the efficiency of fuel and energy companies.

**Keywords:** algorithm; data envelopment analysis; CCR model; BCC model; technical system; combined heat and power plant





## 1. Introduction

The significance of developing theoretical and methodological foundations for assessing the effectiveness of complex systems emerges from the need to modernize the management system of modern enterprises. Such changes are necessary due to the requirements of a rapidly changing external environment. This forms a set of new approaches to understanding the quality of an organization's activities, and consequently its viability. This becomes possible if the main criterion is met: making decisions relevant to certain conditions in which the reproduction process of the objects in the system is carried out [1–5]. Therefore, this article will discuss in detail how, depending on the goal of the decision maker, the conditions for setting the task and the results of calculating performance indicators change.

Many scientists [6–10] have addressed the problem of evaluating effectiveness. In [11], we presented a comparative analysis of all the methods of evaluating effectiveness. Among the methods analyzed was the data envelopment analysis (DEA) method, which is one of the most effective methods for evaluating the effectiveness of complex systems [12–15].

Scientists from different countries have proposed various modifications of the basic models in order to improve the DEA method [16–19]. Research in this direction is still ongoing. New publications with practical applications of the method in various fields are constantly appearing [20–24]. New DEA models are also emerging to solve the problems of evaluating the effectiveness of complex systems [25–27]. Most of the theoretical

and methodological features have been described in various scientific articles and textbooks [28–31]. In addition, scientists from different countries and scientific groups have presented various classifications of the methodology and its description. One of the method founders, William W. Cooper, created a website to highlight the development of the theoretical and practical foundations of the DEA method, as well as various innovations in the practical application of the method [32]. Furthermore, some scientists conducting research in this direction are seeking to popularize the method in scientific circles, having published textbooks, monographs, articles, and operating websites with basic information about the DEA method. There are several main theoretical sources of scientists from different countries on the models of the method and their application [33–35]. In addition, a conference dedicated to scientific research on the DEA method has been held [36]. All these sources contain various scientific works devoted to this method and the peculiarities of its application. Understanding the variety of the uses of the features, models, and modifications of basic models is somewhat complicated, with much time being required to study all the submitted works, each of which describes new scientific developments and additions to the method.

In modern scientific works, the DEA method is primarily considered when solving a variety of practical technical and economic problems [36–40]. On the basis of the practical applications, various methodologies are identified with modified models of the DEA method being used. The latest developments in this direction are described in various scientific sources [41–43]. Recently, modifications of some models have also been presented to solve certain practical problems in various industries [44–46]. Despite the interest of certain scientific groups, the popularization of the method, publications, and new developments in this field, the DEA method is still in the formation stage. Therefore, this article, to some extent, fills a gap in the scientific literature on the analysis of the effectiveness and productivity of complex systems based on the DEA method. In addition, it describes a new algorithm for applying the method in the fuel and energy complex, where the DEA method has not previously been used.

New scientific publications from different countries [47–51] have been devoted to increasing efficiency in the fuel and energy complex, focusing mainly on the introduction of new technologies and modernization of equipment for combined heat and power plants (CHPPs). The use of improved types of fuel for CHPPs is also proposed [52,53]. The efficiency of heat supply system facilities is assessed based on the performance specifications provided in quality standards and rules. Using the DEA method to improve efficiency in this area has not been presented. Therefore, we find it relevant and interesting to develop an algorithm for using the DEA method in fuel and energy complex enterprises, applying it to assess the efficiency of combined heat and power plants. This algorithm is based on DEA models.

This research seeks to solve the problem of increasing the efficiency of enterprises in the fuel and energy complex. Because such enterprises are complex systems, it is difficult to optimize their work, taking into account all the technical indicators of such enterprises. The study proposes to solve this problem using the DEA method. We modelled the cost and result indicators to achieve system efficiency. We are convinced that the algorithm and experimental results presented will allow us to apply the DEA method for the first time to improve the efficiency of fuel and energy companies.

In the continuation of the study, the methodology, the data obtained, and the conclusions from the experiments presented will form the basis of the decisions support system algorithm for the combined heat and power plant.

This paper is organized as follows: Section 1 introduces the work. Section 2 describes the basic concept of the DEA method and the basic models of the method. Section 3 then provides an overview of the main publications on the use of the DEA method, presenting a comparative analysis of the basic models and features of the use of the DEA method. The proposed methodology for applying the DEA method to fuel and energy complex enterprises is detailed in Section 4, while Section 5 presents the results of the experiments

on the use of the methodology to improve the efficiency of the CHPP. Finally, Section 6 presents our main conclusions.

## 2. Basic Concepts

The significance of developing theoretical and methodological foundations for assessing the effectiveness of complex systems emerges from the need to modernize the management system of modern enterprises. Such changes are necessary due to the requirements of a rapidly changing external environment. The management concept, which orients an organization of any type towards the effectiveness of its activities, requires the constant improvement of the existing methods and tools to enhance the efficiency of modern enterprises.

Farrell [54] was the first to propose measuring the efficiency of a complex system based on one input and one output indicator. In 1957, the author applied this model to the elements of the United States agricultural system, failing, however, to combine all the possible input and output indicators to calculate the effectiveness of the elements studied. This methodology was developed by Charnes, Cooper, and Rhodes in 1978 [55]. They suggested using mathematical programming to solve this kind of problem. The DEA method was then formulated. Over the years, the method has been widely applied in various fields to calculate both the technical and economic efficiency of the functioning of enterprises as complex systems [56–59].

At this time, the basic models of the DEA method were formulated [51]. These models have become widespread in research around the world with practical applications in various fields. Data envelopment analysis is a mathematical programming method for determining relatively useful actions of decision-making units and technical efficiency [53]. DEA helps enable the use of multiple inputs and outputs in a linear software model that calculates efficiency [53].

The system for which comparative efficiency is evaluated is called the decision-making unit (DMU) [58]. A DMU uses certain resources at the input and converts them into products of a specific type at the output. For example, it can be proposed that the DMUs under study are a CHPP. Here, it is worth clarifying the type of enterprise under consideration. This article does not consider thermal power plants designed to produce electricity for industrial purposes for external consumers but concerns thermal power plants designed for the production of heat for industrial purposes, that is, for district heating.

The combined heat and power plants considered in this article have as their main production function, like boiler houses, the production of heat (hereinafter in the calculations of the supply of thermal energy to heat networks), which is supplied to the grid for consumers. Electricity generation is an auxiliary function of such combined heat and power plants. The electricity generated is produced in relatively small quantities and is only used for the specific needs of the combined heat and power plants. Therefore, the main objective of the decision maker in the control of the CHPP is to regulate the heat supplied to the network for consumers at different times of the year. Increasing the electricity production is often impractical for such CHPPs, as it requires the construction of additional infrastructure for the consumers of the generated electricity. Therefore, we do not consider the electricity generation of these CHPPs in this study, as this function of the CHPPs under study is not the main one. Electricity is consumed only by the CHPPs themselves for their own needs.

Heat generation is the main function of such plants and satisfies the heating needs of consumers in urban agglomerates with difficult climatic conditions in cold periods. Therefore, we consider heat production as the main output of CHPPs, requiring adjustments in warm and cold seasons. As input, it uses the available thermal power of the equipment and the consumption of conventional fuel for the released fuel cell. And, as a result of the work, it has as output: the production of heat (hereinafter in the calculation of supply of thermal energy to heat networks).

In 1957, Farrell considered measuring efficiency in the presence of multiple inputs and outputs by assigning weights to inputs and outputs [54].

In this analysis, the fundamental point is efficiency, which is generally defined as the quotient of dividing the sum of all output parameters by the sum of all input factors [54].

In general, the formula is as follows [56]:

$$Te = \frac{\sum weighted\ output\ parameters}{\sum weighted\ input\ parameters} \tag{1}$$

where Te is technical efficiency.

Thus, efficiency according to the DEA method is generally calculated as the ratio of the sum of the weighted results of the enterprise's activities (DMU) to the sum of the weighted funds used.

For all DMUs of the studied sample of any Pareto-efficiency system, it is impossible to increase one of the outputs (decrease one of the inputs) so as not to decrease (increase) the other as a result [55].

Accordingly, when considering different tasks in improving the efficiency of the system's enterprises, different models of the DEA method are used. There are a considerable number of models. Therefore, when choosing a model of the DEA method in classical theory, the focus is typically on 3 main aspects that must be considered when constructing the DEA model [56]:

1. What input and output indicators of the model will be used and in what quantity, to solve the goal.
2. What determines the choice of a constant or variable scale effect: constant return to scale or variable return to scale.
3. What determines the choice of the model orientation: output-oriented model or input-oriented model.

In practice, there may be more aspects. We will consider all the features of the model selection in more detail in the following sections of this article.

### 2.1. The CCR Model

The founders of the model are Charnes, Cooper, and Rhodes in 1978 [55], with the model, thus, being called the CCR model, as an abbreviation of the authors' surnames. It was one of the first to be developed, being considered a basic model in the scientific literature. Based on this model, the DEA method has been supplemented with many new models and has been further developed.

The CCR model was based on the Farrell method of measuring the efficiency of the units studied using production functions [34]. The equation with a nonlinear Farrell program displays a simple production ratio with a single input and output parameter [12–14]. In this model, multiple input and output parameters of each enterprise are combined into various scalar input and output parameters. This model is based on a constant return to scale (CRS).

It can be described by a formula that represents a solution to the maximization problem:

$$Ef = \frac{\sum_{j=1}^{s} U_j Y_{j_o}}{\sum_{i=1}^{r} V_i X_{i_0}} \rightarrow max! \tag{2}$$

Under the following restrictions:

$$Ef = \frac{\sum_{r=1}^{s} U_r Y_{r_m}}{\sum_{i=1}^{m} V_i X_{i_m}} \leq 1$$

$m = 1, 2,..., $ n, $U_r \geq 0$, $j = 1, 2,..., $ s, $V_i \geq 0$, $i = 1, 2,..., $ r.
where
*Ef* is the efficiency value of the enterprise.
*n* is the number of units compared.
*r* is the number of input factors.

$s$ is the number of output parameters.

$x_{i_0}$ is an observed indicator of the i-th input factor of the enterprise.

$Y_{i_0}$ is an observed indicator of the j-th output parameter of the enterprise.

$X_{i_m}$ is the observed indicator of the i-th input factor of the m-th enterprise with i = 1,..., r and m = 1,..., n.

$Y_{i_m}$ is the observed indicator of the j-th output parameter of the m-th enterprise with i = 1,..., r and m = 1,..., n.

$U_r$ is the weighting of output factor i with I = 1,..., r.

$V_i$ is the weighting of input parameter j with j = 1,..., s.

When solving this maximization problem, the problem of having a quotient in the division of two linear aggregated values arises. These maximization problems are also called linear quotient programming. There are also many possibilities for transforming the linear programming of the quotient into a linear programming problem. For this purpose, Charnes, Cooper, and Rhodes, in 1978, formed the so-called inverse program of the following function [55]:

$$Ef = \frac{\sum_{i=1}^{r} V_i X_{i_0}}{\sum_{j=1}^{s} U_j Y_{j_o}} \to min! \tag{3}$$

Subject to the following conditions:

$$Ef = \frac{\sum_{i=1}^{m} V_i X_{i_m}}{\sum_{r=1}^{s} U_r Y_{r_m}} \leq 1$$

where m = 1, 2,..., n, $U_r \geq 0, j = 1, 2,..., s, V_i \geq 0, i = 1, 2,..., r$.

### 2.2. The BCC Model

A significant disadvantage of the CCR model discussed above is the premise of linear homogeneity [56]. Banker, Charnes, and Cooper in 1984 [13] developed a model that eliminates this drawback. It was named the BCC model, after the authors' surnames. These models differ from CCR models by adopting variable returns to scale (VRS). Variations in the BCC model make it possible to identify an increasing or decreasing scale effect for each object [60].

For the above-described linear production models, this means that if the production object $(X_0, Y_0)$ is valid under the given economic conditions, then the object $(tX_0, tY_0)$ where we have the number $t > 0$ will also be valid.

Since the concept of scale effect is important in mathematical and practical economics, we focus on it in a little more detail. It is believed that for a specific object (a point in the socio-economic space) there are the following [61]:

- A constant scale effect, if the limiting coefficient of the ratio of the relative increment of the output vector to the relative increment of the cost vector is equal to one;
- An increasing scale effect, if the limiting coefficient of the ratio of the relative increment of the output vector to the relative increment of the cost vector is greater than one;
- A decreasing scale effect, if the limiting coefficient of the ratio of the relative increment of the output vector to the relative increment of the cost vector is less than one.

A mathematical representation of the variable scale effect can be produced by adding a new variable $U_0$ to the objective function of the original model.

$$E_0 = \frac{\sum_{j=1}^{s} U_j Y_{j_0} + U_0}{\sum_{i=1}^{r} V_i X_{i_o}} \to max! \tag{4}$$

Subject to the following conditions:

$\frac{\sum_{j=1}^{s} U_j Y_{jm} + U_0}{\sum_{i=1}^{r} V_i X_{im}} \leq 1, U_j, V_i \geq 0.$

$U_0$ is the scale effect, $U_0 > 0, U_0 < 0, U_0 = 0$.

At the same time

$U_0 > 0$ is the increasing return to scale;

$U_0 < 0$ is the decreasing return to scale;

$U_0 = 0$ is the constant return to scale.

The equation presented (4) is the BCC-Output Model.

Now, let us imagine a multiplier primal BCC input model with variable slacks [62].

$$max\sum\nolimits_{j=1}^{s} \mu_j y_{j0} + U_0 \qquad (5)$$

Subject to the following conditions:

$\sum_{i=1}^{r} T_i X_{i0} = 1$

$\sum_{j=1}^{s} \mu_j y_{j0} - \sum_{i=1}^{r} T_i X_{i0} + \sum_{m=1}^{n} U_0 \leq 0$

for all studied objects m = 1,..., n; $\mu_j t_j \geq 0$; $U_0$ is free.

The fractional program is converted to an LP problem using the Charnes and Cooper transformation and the Multiplier BCC Output model is as below [60].

$$\frac{1}{E_0} = min\sum\nolimits_{i=1}^{m} V_i X_{i_0} + V_0 \qquad (6)$$

Subject to the following conditions:

$\sum_{r=1}^{s} U_r Y_{r_0} = 1$

$\sum_{i=1}^{m} V_i X_{ji} + V_0 - \sum_{r=1}^{s} U_r Y_{r_j} \geq 0$

j = 1, 2... *n*, $U_r V_i \geq \varepsilon$, r = 1, 2..... s, i = 1, 2...., m, $V_0$ is free (unrestricted in sign).

### 2.3. The Combined Heat and Power Plant

The research is designed to evaluate the efficiency of enterprises in the fuel and energy complex. Namely, the study is carried out for combined heat and power plants (CHPPs) of the heating system. Therefore, a description of the CHPPs will be presented below.

A combined heat and power plant is an enterprise whose products are electricity, as well as heat released in the form of steam and hot water, and the "raw material" is organic fuel (coal, gas, fuel oil, etc.) [49]. The equipment of the power plant is used for the economic conversion of chemical energy into electrical energy. Steam is supplied directly for industrial and household needs or is partially used for preheating water in special boilers (heaters), from which water is sent through the heating network to consumers of thermal energy. The steam condensate given to the thermal consumer is returned to the CHPP by a reverse condensate pump.

The connection of the CHPP to other stations of the power system is carried out at high voltage through step-up transformers.

Below is the composition of the combined heat and power plant:

- Fuel economy and fuel preparation system;
- Boiler plant: a combination of the boiler itself and auxiliary equipment;
- Turbine plant: steam turbine and its auxiliary equipment;
- Installation of water treatment and condensate treatment;
- Technical water supply system;
- Ash removal system (for CHP plants operating on solid fuel);
- Electrotechnical equipment and electrical equipment control system.

### 3. Related Work

This article considers in detail two main basic models: the CCR model and the BCC model. They are partially linear models. Scientists from different countries have developed a number of basic DEA models that have been applied in certain areas. For example, the additive DEA model (ADD) was proposed by Charnes, Cooper, Golany, Seiford, and Sturz [16] in 1985. This model does not have an input or output orientation. The ADD function of the model determines the efficiency identified from the reserve variables [59].

The Slack Based Model (SBM) is a continuation of the additive model and was introduced by Tone in 2001 [34]. It differs from the ADD model in that it is already a relative measure, not an absolute one. The Multiplicative Model was described by Charnes, Cooper, Seiford, and Sturz [16] for Cobb and Douglas production functions or functions that are partially linear-logarithmic. Thus, the efficiency frontier has a piecewise logarithmic frontier, and not a piecewise linear one, as in the case of CCR, BCC, and other models. Additionally, virtual inputs and outputs are formed multiplicatively, not additively.

### 3.1. Comparison of Characteristics and Criteria When Choosing the Main DEA Models: CCR and BCC

As a summary of the basic models of CCR and BCC, we present the following comparative scheme, which includes the main characteristics of these models of DEA. Figure 1 below demonstrates the comparative overview of the basic DEA CCR and BCC models.

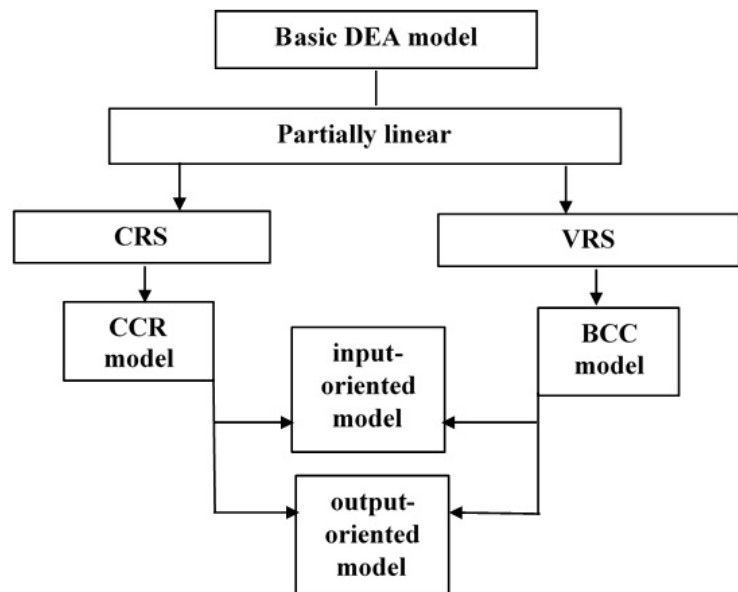

**Figure 1.** Comparative overview of the basic DEA CCR and BCC models.

Thus, the basic DEA models can be classified according to the following criteria [20]:

- Type of production function;
- Selected orientation (input- or output-oriented models, as well as models without orientation);
- Acceptance of a constant (CRS) or variable return to scale (VRS).

As described above, CCR and BCC models are models with orientation to either the input or the output. The constant scale effect is applied to the model CCR. In BCC models, variable economies of scale are considered. The CCR and BCC models are partially linear models.

The BCC model is useful for variable returns to scale and measures only the net technical efficiency of each DMU [60].

The CCR model studied above considers the aggregate (technical and scale) efficiency, whereas the BCC model decomposes the total aggregate efficiency of a unit into its purely technical and scale efficiency.

To evaluate production functions with the help of DEA, it is especially important to choose a suitable DEA model for each specific case. Table 1 shows a comparison of the CCR and BCC models.

**Table 1.** Comparison of the characteristics of the described CCR and BCC models.

| Model | Scale Effect | Range Efficiency | Type of Efficiency | Data | |
|---|---|---|---|---|---|
| | | | | x | y |
| CCR-Output | CRS | [0.1] | TE | S-p | U-s |
| CCR-Input | CRS | [0.1] | TE | S-p | U-s |
| BCC-Output | VRS | [0.1] | TE | U-s | S-p |
| BCC-Input | VRS | [0.1] | TE | S-p | U-s |

CRS—Constant Return to Scale, VRS—Variable Return to Scale, TE—Technical Efficiency, U-s—unrestricted in sign, S-p—Semi-p (= semi-positive)—only positive values are allowed.

Thus, Table 1 shows that the CCR and BCC models are used to measure technical efficiency. For all basic models, efficiency is measured in the range from 0 to 1.

In scientific articles, CCR and BCC models have a wide range of practical applications. They have been used to evaluate the efficiency of enterprises in economics, banking, agriculture, medicine, industry, trade, transport, and many others. However, the use of DEA models in the fuel and energy complex has not been investigated.

We next take a closer look at the main characteristics that need to be considered when choosing a DEA model.

*3.2. Comparison of Characteristics for the DEA Method*

When choosing a DEA model, the first decision is whether to use a constant scale effect (CRS) or variable scale effect (VRS). Upon closer examination, it can be said that the output parameters change proportionally to the input parameters during the CRS. When the VRS changes the output parameters, the output parameters may change disproportionately [60].

Färe, Grosskopf, and Levell [63] divide technical efficiency into two related multiplier components. The division is based on the dependence of efficiency on the magnitude of the scale. Considering enterprises for returns on scale, we can speak of the dependence of productivity on an increase in the resources used for production. If the increase in productivity is proportional to the increase in the number of resources used, then the enterprise has a CRS.

An enterprise achieves the highest possible level of productivity if it is 100% efficient, with both a constant and a variable return on scale.

If an enterprise is 100% efficient with VRS, but fails to achieve full efficiency with CRS, there then arises the efficiency depending on the magnitude of scale—scale efficiency (SE).

The formula of this regularity is as follows:

$$\text{TE} = \text{SE} \times \text{PTE} \tag{7}$$

where

TE—technical efficiency with constant return of scale;

PTE—technical efficiency with variable return of scale, or pure technical efficiency;

SE—efficiency depending on the magnitude of the scale.

When calculating these performance indicators, it is possible to identify the sources of inefficiency. In the case of PTE, inefficiency may be caused by the inefficient operation of the enterprise. Then, in the case of SE, inefficiency can be caused by unfavorable conditions. In the experimental section, we will show this with examples.

Figure 2 below demonstrates the difference between the three performance indicators (TE, SE, and PTE) discussed above.

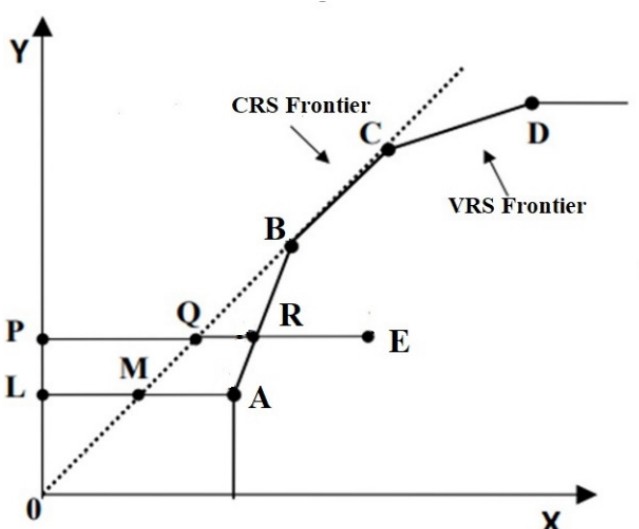

**Figure 2.** Efficiency depending on the magnitude of the scale [8].

In Figure 2, the input parameters are located along the *X*-axis. The *Y*-axis shows the output parameters.

The direct line is a line on which 100% efficient enterprises with a constant return of scale should be located. On the F curve, there are 100% efficient enterprises with variable returns of scale. Thus, enterprise A, lying on the curve F, is fully efficient with VRS (PTE = 1), but not efficient enough with CRS. Therefore, the efficiency, depending on the magnitude of the scale, is determined by the ratio [64]

$$SE(A) = \frac{LM}{LA} < 1 \tag{8}$$

Both enterprise B and enterprise C are 100% effective at both CRS and VRS, so they operate at the highest possible level of productivity. For an inefficient enterprise E, efficiency can be determined using the following formula:

$$SE(E) = \frac{PQ}{PE} = \frac{PE}{PR} \tag{9}$$

Figure 2 and the calculations show that the inefficiency of enterprise E is caused by both inefficient production and unfavorable conditions.

**4. Materials and Methods of using DEA Models to Calculate the Efficiency of Fuel and Energy Complex Enterprises**

In the previous paragraphs, the basic DEA models were investigated. We now consider the possibility of applying these models in practice in a specific area and build an algorithm for using DEA models to calculate the efficiency of fuel and energy complex enterprises. To do this, we will follow the next steps for evaluating the effectiveness of the DEA method:

1. Studying the scope in which it is planned to evaluate the effectiveness of the studied objects (CHPP);
2. Setting a goal;
3. Selection of the investigated CHPP;
4. Analysis of the external environment and basic CHPP operations;
5. Selection of input and output parameters (evaluation factors);
6. Selection of the model and its orientation in accordance with the goal;
7. Software selection;
8. Interpretation of the results obtained;
9. Recommendations for improving efficiency.

Based on these steps, we have compiled a diagram of an algorithm for evaluating the effectiveness of the DEA method. It is shown in Figure 3.

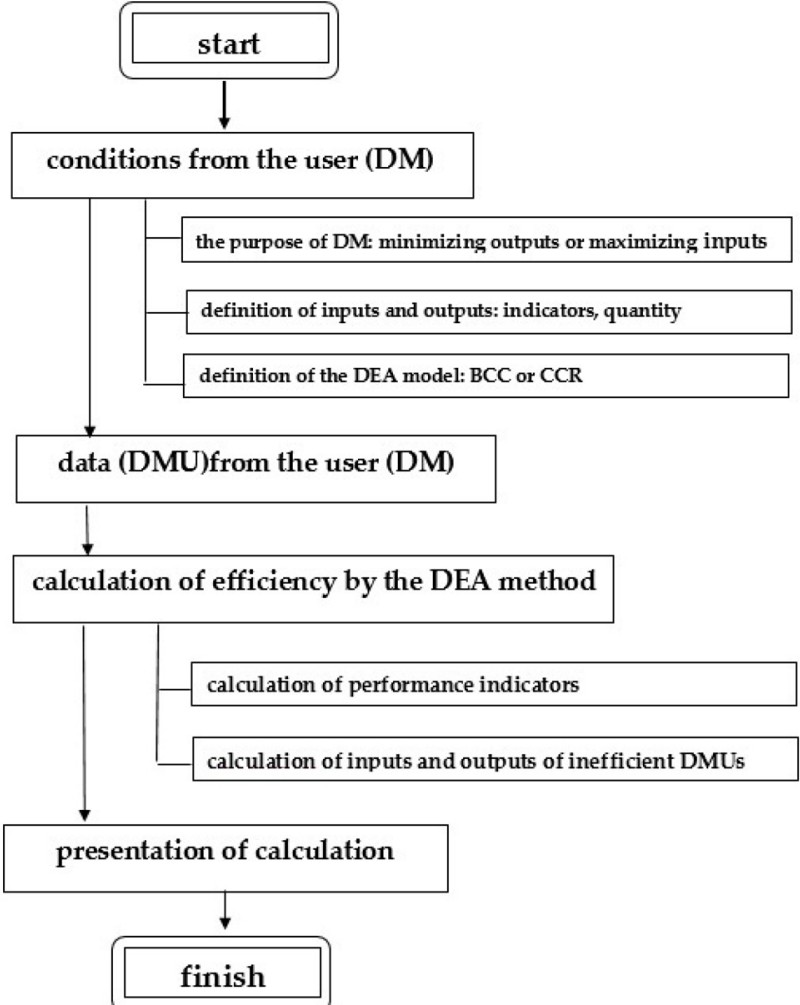

**Figure 3.** Scheme of the algorithm for evaluating the effectiveness of the DEA method.

This algorithm can be applied to many complex systems. In this article, we have applied it to a complex technical system. This is justified by the fact that in the management of technical systems, decision makers often face the difficulties of regulating constituent objects while improving the efficiency of the system as a whole. The measurement of a large number of system performance indicators, as well as their regulation, often creates problems in the management of the system as a whole. We applied the algorithm in the district heating system. This is justified by the global limitation of energy and heat resources across the world. This system, in our opinion, is subject to the emergence of a number of problems in this regard, as, in accordance with global strategies, energy and heat should be saved. The district heating system, in our view, requires the application of the efficiency-improvement method, which has hitherto not been applied in this system as it will contribute to the reduction of resource consumption and increase the efficiency of enterprises in the system. The experimentation of the algorithm proposed in the article is carried out for CHPPs in the district heating system. The following reasons underlie the choice of these enterprises and the system itself:

- The global strategy of scientific and technological development of fuel and energy companies;
- The need to modernize the management system of modern fuel and energy companies in accordance with the requirements of a rapidly changing external environment;

- The aim of increasing competitiveness in enterprises;
- The transition of fuel and energy companies to advanced digital, intelligent production technologies.

In addition, one of the most important problems for the fuel and energy sector today is that of ecology and environmental pollution. For more than a decade, residents of cities where combined heat and power plants are located have been forced to bear unfavorable living conditions associated with pollution. Such emissions also cause damage to the global environment as a whole. The global strategy of conservation and reduction of environmental pollution dictates the policy of the world powers to maximize the production of heat and electricity with minimal damage to the planet.

As noted above, the main methodologies recently proposed by scientists have focused on the introduction of new technologies and modernization of equipment for combined heat and power plants. The use of improved types of fuel for CHPPs is also proposed. The efficiency of the heat supply system facilities is assessed based on the performance specifications specified in the quality standards and rules. The algorithm proposed in the article for using the DEA method is based on a different approach, which does not involve the replacement of equipment or technology. Therefore, it requires no additional high financial investment. The proposed methodology allows us to consider only external indicators (input and output indicators) to assess effectiveness. This simplifies calculations, unlike the situation of joint analysis of internal and external indicators. The proposed algorithm allows for the flexibility and high adaptability of the system in accordance with changes in environmental factors and changes in the purpose of the decision maker. Using the DEA method allows us not only to evaluate the effectiveness of the enterprises analyzed, but also to configure the inputs and outputs of non-effective enterprises to achieve efficiency.

Below, we describe this algorithm in more detail for this field of application.

Next, in the experimental part, we will consider six experiments. In these experiments, we use the CCR model and the BCC model. In the first four experiments, we will use the following input and output data:

(1) Available thermal power of the equipment (Gcal/hour)—input ($x1$).
(2) The consumption of conventional fuel for the released fuel cell released (thousand tons/year)—input ($x2$)
(3) Supply of thermal energy to heat networks (thousand Gcal)—output ($y1$).

In experiment №5, we will use the following inputs and outputs:

(1) The available thermal power of the equipment (Gcal/hour)—input ($x1$).
(2) The consumption of conventional fuel per released fuel cell (thousand tons/year)—input ($x2$).
(3) Supply of thermal energy to heat networks (thousand Gcal)—output ($y1$).
(4) The mass of the emission (thousand tons per year)—output ($y2$).

In experiment №6, we used the following inputs and outputs:

(1) Heating capacity of turbo generators (Gcal/hour)—input ($x1$).
(2) Power of peak hot water boilers (Gcal/hour)—input ($x2$).
(3) Power of steam boilers (Gcal/hour)—input ($x3$).
(4) Consumption of conventional fuel for the released fuel (thousand tons/year)—input ($x4$).
(5) Heat supply to the grid (thousand Gcal)—output ($y1$).

### 4.1. Selecting the Main Criteria for Assessing Effectiveness

For this study, the operation of the district heating system and the technological processes of the CHPP plant were studied.

The main output indicator of the CHPP is the release of thermal energy to heat networks [20]. Taking into account the environmental factor, which was described above, it is reasonable to consider also an output indicator: the mass of emissions. However, if the inclusion of the environmental aspect is not required for the decision-maker (DM) when

achieving a specific goal, then the model can be built with one input indicator: the release of thermal energy into the network.

Thus, as input indicators in this study, we use the following:

(1)　The available thermal power of the equipment (Gcal/hour).
(2)　The consumption of conventional fuel per released fuel cell (thousand tons/year).

As output indicators, we use the following:

(1)　Supply of thermal energy to heat networks (thousand Gcal).
(2)　The mass of the emission (thousand tons per year).

There may be situations when there is a need to reduce part of the output indicators, that is, it is necessary to reduce the mass of emissions from boilers. To do this, for calculations, we do not use the actual values of the output indicators but their deviations from the threshold value set at a level obviously exceeding the values of the corresponding indicators for all objects studied.

Furthermore, in order to see how to regulate the available thermal power for the main types of equipment, it is more expedient to divide it into power for each type of main equipment. Therefore, this indicator is divided into the following three indicators:

- Heating capacity of turbo generators (Gcal/hour);
- Capacity of peak hot water boilers (Gcal/hour);
- Power of steam boilers (Gcal/hour).

In this study, we conduct experiments on various variations of input and output indicators.

It should be noted that DEA models have limitations on the number of observations to obtain reliable calculated data. This is reflected in many scientific studies. Accordingly, in the literature, it is accepted that the number of observations is calculated using the following formula [64]:

$$N > max\{M \; x \; S, 3(M+S)\} \tag{10}$$

where

$N$ is the number of CHPs, $M$ is the number of input indicators, and $S$ is the number of output indicators.

Therefore, to conduct experiments, it is advisable to choose the required number of observations, no fewer than what is obtained for this sample when calculating according to the formula presented above [64].

In a real calculation situation, if the number of observations is lower than that determined by the formula, this problem can be solved in two ways. For example, this can be achieved by increasing the number of CHPPs. If it is impossible to increase the number of CHPPs investigated, then the scientific literature describes the possibility of using observations on the same CHPPs in a different time period, taking them as new CHPPs [64].

Depending on the goals of the decision maker, the orientation of the DEA model may be different. The model can be input-oriented if the goal is to reduce the input parameters, and if the goal is to increase output indicators, then we use an output-oriented model.

In our study, we conduct experiments on two types of orientation: input and output. Thus, we can see the difference and draw conclusions about the feasibility of each option.

We perform experiments using the main models: CCR and BCC. We conduct experiments with both a constant scale effect and a variable one to see if there are any matches on the effectiveness of the CHPP from the analyzed sample at VRS and CRS. We, thus, see what causes the inefficiency of the enterprises studied: inefficient work and/or unfavorable conditions.

We have chosen the Charnes, Cooper, and Rhodes model and the Banker, Charnes, and Cooper model because these models are the basic models of the DEA method. Since these models are the basis for the operation and use of the DEA method. In order to use additional DEA models, we first need to analyse the effectiveness of the basic models. We have chosen these models as models of widespread use. Often, the models discussed in this

article are sufficient to analyse the efficiency of technical systems. We have chosen these models so that readers can understand the essence of the DEA method. And, in the future, they can also try to use additional DEA models and easily analyse using the basic models.

*4.2. Calculating, Interpreting Results, and Making Recommendations to Improve Efficiency*

We now describe the software for conducting experiments. There is currently a large number of software products based on DEA models. The most common include DEEP, Master, DEAD Frontier, PIM-DEA Soft, DEAS, OpenDEA, DEAOS, EMS, DEA Solver Online, Frontier Analyst, DEA-SolverPro, and KonSi—Data coverage analysis for comparative analysis. A comparative study of such software was conducted by Iliyasu et al. [53].

Our review of all these software capabilities led to the selection of DEEP as the most suitable software for research on CCR and BCC models, being that used in this study [65].

Nonetheless, the existing software products are unfortunately limited only to assessing the current level of efficiency, without offering means to explain this level, issuing recommendations on ways to achieve the required level of efficiency (in particular, by redistributing resources between subsystems).

When using the calculated results, the question of the interpretation of the results obtained arises, as well as the development of recommendations for improving efficiency in this field. In addition, it is difficult to use these software tools without a detailed study of the DEA method and the software support itself. Existing software products do not use "large" database management systems—Oracle, PostgreSQL, MySQL, etc. The data is stored in Excel tables, in a Microsoft Access (mdb) database, and in text files. Accordingly, engineers and other decision makers often cannot correctly use the models of the DEA method, using the presented software each time to solve a specific goal. Accordingly, the actual use of such software in practice for solving problems in complex technical systems is limited. For specialists in the field of computer technology, solving problems in complex technical systems is difficult without specialist knowledge and experience in the field under study. It is also difficult for them to correctly form input and output parameters, as well as the model itself, its orientation and scale, as the technical systems are complex. In order to be able to evaluate their effectiveness, a detailed study of such systems functioning is required. In addition, recommendations should also contain information on how, and according to what, input and output indicators can be changed, as the program outputs only the optimal combination of input and output values to maximize efficiency in the sample of studied enterprises. It is, thus, difficult to interpret the results obtained. Accordingly, how to achieve the recommended indicators should be described in the expert's recommendations.

This problem may be solved by a programmed DSS at the enterprises studied. Accordingly, to automate the formation of result interpretation when programming a DSS, a large number of preliminary experiments are required.

Therefore, the next section is devoted to conducting such experiments and analyzing the data obtained.

## 5. Results

We now present the experimental part of the study. In this section, we test the methodology given in the previous paragraph for using basic DEA models to calculate the efficiency of fuel and energy complex enterprises. We investigate changes in the sample indicators and changes in models, their orientation, inputs, and outputs when achieving different goals of the DM. The objects of the study will be combined heat and power plants. A total of 27 CHPPs will be represented in the sample [66]. This is a sufficient number of CHPPs to conduct experiments and ensure the correctness of calculations using the DEA method. In the presentation of research results, in order to simplify the visualization of the data obtained, only a part of the sample is presented. We next present the results for six CHPPs in order to describe and analyze the main results and patterns obtained in the experiments.

To build a basic model, for initial experiments, we take the simplest combination of inputs and outputs: two inputs and one output to demonstrate the possibility of testing this method in the field studied. Let us imagine the input and output parameters.

Inputs:

(1)    Available thermal power of the equipment (Gcal/hour)—input (x1).
(2)    The consumption of conventional fuel for the fuel cell released (thousand tons/year)—input (x2).

Outputs:

(1)    Supply of thermal energy to heat networks (thousand Gcal)—output (y1).

The initial data are presented in Table 2.

**Table 2.** Initial data on two input indicators and one output indicator of the studied CHPP sample.

| № CHPP | O (y1) | I (x1) | I (x2) |
|---|---|---|---|
| 1 | 3635 | 1554 | 642 |
| 2 | 4519 | 1530 | 790 |
| 3 | 1077 | 752 | 401 |
| 4 | 3626 | 1554 | 641 |
| 5 | 4380 | 1405 | 767 |
| 6 | 1001 | 752 | 386 |

O—output; I—input.

If it is important for the decision maker to increase the supply of thermal energy to heat networks, it is advisable to use the output-oriented DEA model. If it is important for the decision maker to reduce the consumption of conventional fuel for the released fuel cell and reduce the available thermal power of the equipment, then it is advisable to use the input-oriented DEA model.

We next conduct experiments on different conditions and goals for the decision maker. We calculate the efficiency using the DEA method in DEEP software [67] (http://www.uq.edu.au/economics/cepa/deap.php (accessed on 22 May 2022)).

*5.1. Experiment 1—The Output-Oriented CCR Model with Two Input and One Output Parameters*

Next, we build a basic output-oriented CCR model. The goal for this experiment is to increase the supply of thermal energy to heat networks.

The results obtained are presented in Table 3.

**Table 3.** Efficiency calculation based on the output-oriented CCR model with two input and one output parameters.

| № CHPP | Eff | O (y1) | I (x1) | I (x2) |
|---|---|---|---|---|
| 1 | 0.873 | 4161.898 | 1554 | 642 |
| 2 | 0.978 | 4620.025 | 1530 | 790 |
| 3 | 0.466 | 2312.812 | 752 | 401 |
| 4 | 0.872 | 4158.590 | 1554 | 641 |
| 5 | 1.000 | 4380.000 | 1405 | 767 |
| 6 | 0.442 | 2263.190 | 752 | 386 |

Eff—effectiveness.

It can be seen that only CHPP 5 has an efficiency indicator of 1 and is, thus, effective. It does not require setting the output indicator. Therefore, in the table, the output indicator y1 remains unchanged for CHPP 5. For all other CHPPs, the y1 indicator changes. Table 3 shows the y1 values required to achieve the effectiveness of this CHPP. For example, the efficiency indicator calculated by the DEA method for CHP1 is 0.873 with the maximum possible being 1. The input indicator output (y1) is initially equal to 3635.

In order to increase the efficiency of this CHPP1 to 1, and so make the enterprise fall on the efficiency frontier built on this sample, it is necessary to increase output (y1) to 4161,898. The efficiency coefficient calculated by the DEA method will be equal to 1, and, accordingly, the enterprise will become efficient.

Input indicators input (x1) and input (x2) remain unchanged when the model is oriented to the output of the CCR model; only the output indicator changes.

*5.2. Experiment 2—The Input-Oriented CCR Model with Two Input and One Output Parameters*

We now build a basic input-oriented CCR model. The goal for this experiment is to reduce the consumption of conventional fuel for the released fuel cell and reduce the available thermal power of the equipment. The results obtained are presented in Table 4.

**Table 4.** Efficiency calculation based on the input-oriented CCR model with two input and one output parameters.

| № CHPP | Eff | O (y1) | I (x1) | I (x2) |
|--------|-------|--------|----------|---------|
| 1 | 0.873 | 3635 | 1357.263 | 560.723 |
| 2 | 0.978 | 4519 | 1496.544 | 772.725 |
| 3 | 0.466 | 1077 | 350.182 | 186.732 |
| 4 | 0.872 | 3626 | 1354.980 | 558.907 |
| 5 | 1.000 | 4380 | 1405.000 | 767.000 |
| 6 | 0.442 | 1001 | 332.607 | 170.726 |

From this table it can be seen that the performance indicators have not changed relative to experiment 1. In this case, only the orientation of the model has changed. Now the DEA model minimizes the input metrics without changing the output metric.

As can be seen from Table 4, in experiment 1, only CHPP 5 has an efficiency index of 1 and is, thus, effective. It does not require setting the output indicator. Therefore, in the table, the indicators of inputs (x1) and (x2) remain unchanged for CHPP 5. For all the other CHPPs, the values of inputs (x1) and (x2) change. Table 4 shows the values (x1) and (x2) required to achieve the effectiveness of this CHPP. For example, the efficiency indicator calculated using the DEA method for CHPP 1 is 0.873 with the maximum possible being 1. The input indicator (x1) is initially equal to 1554 and the input indicator (x2) is initially equal to 642. In order to increase the efficiency of this CHPP 1 to 1, and so make the enterprise fall on the efficiency frontier built on this sample, it is necessary to reduce input (x1) to 1357.263; input (x2) should be reduced to 560.723. In this case, the efficiency coefficient calculated using the DEA method will be equal to 1 and, accordingly, the company will become efficient.

The output indicator (y1) remains unchanged when oriented to the input of the CCR model. Only the indicators of the inputs change.

*5.3. Experiment 3—The Input-Oriented BCC Model with Two Input and One Output Parameters*

We now build a basic input-oriented BCC model. The goal for this experiment is to reduce the consumption of conventional fuel for the fuel cell released and reduce the available thermal power of the equipment. However, as described above, the BCC model is a VRS model. We make calculations using the same initial data from Table 3 as for the CCR

model. Accordingly, we will see how the performance indicators will change and what the scale effect will be for each CHPP.

The results obtained are presented in Table 5.

**Table 5.** Efficiency calculation based on the input-oriented BCC model with two input and one output parameters.

| № CHPP | Eff | O (y1) | I (x1) | I (x2) |
|--------|-----|--------|--------|--------|
| 1 | 0.893 | 3635 | 1388.020 | 573.429 |
| 2 | 1.000 | 4519 | 1530.000 | 790.000 |
| 3 | 0.761 | 1311 | 572.000 | 292.000 |
| 4 | 0.891 | 3626 | 1385.218 | 571.380 |
| 5 | 1.000 | 4380 | 1405.000 | 767.000 |
| 6 | 0.761 | 1311 | 572.000 | 292.000 |

From this table, it can be seen that, according to this model, 2 enterprises are already effective: CHPPs 2 and 5. They have an efficiency score of 1, and so are effective. They do not require setting up input indicators. Therefore, in Table 5, the values of inputs (x1) and (x2) remain unchanged for CHPPs 2 and 5. For all the other CHPPs, the values of inputs (x1) and (x2) change. Accordingly, Table 5 shows the values of inputs (x1) and (x2) required to achieve the efficiency of the remaining CHPPs. For example, the efficiency indicator calculated using the DEA model BCC method for CHP 1 is 0.893 with the maximum possible being 1. The input indicator input (x1) is initially equal to 1554 and the input indicator input (x2) is initially equal to 642. In order to increase the efficiency of this CHPP1 to 1, and so make the enterprise fall on the efficiency frontier built on this sample, it is necessary to reduce input (x1) to 1388.020. The input (x2) indicator is reduced to 573.429. In this case, the efficiency coefficient calculated using the DEA method will be equal to 1 and, accordingly, the company will become efficient.

The output indicator (y1) according to the BCC model for CHPP 1 has not changed. In this model, unlike the CCR model, the output indicator can also change in the presence of slack movement for the output indicator. In the section describing the models, this phenomenon was described in sufficient detail. Slack movement can be used for both output indicators and input indicators. In our sample, such CHPPs are present, and there is slack movement for both input and output indicators.

Table 6 presents the slack movement for the indicators of the sample studied.

**Table 6.** Slack movement for the indicators of the input-oriented BCC model with two input and one output parameters.

| № CHPP | O (y1) | I (x1) | I (x2) |
|--------|--------|--------|--------|
| 1 | - | - | - |
| 2 | - | - | - |
| 3 | 234 | - | −13.016 |
| 4 | - | - | - |
| 5 | - | - | - |
| 6 | 310 | - | −1.606 |

As we can see from Table 6, slack movement is present for CHPPs 3 and 6. The CHPP data tends towards the CHPP 27 indicators, with CHPPs 3 and 6 tending to the nearest effective frontier point, which for them is CHPP 27. Accordingly, slack movement appears.

As described above, the BCC model, unlike the CCR model, helps to calculate efficiency considering variable return to scale, while determining technical efficiency and scale

efficiency. Therefore, to analyze all performance indicators for this sample, it is advisable to present a comparison of all the performance indicators for the sample studied. Below we present a table comparing performance indicators for the output-oriented CCR and BCC models with two input and one output parameter.

As we can see from Table 7, for this sample, only CHPP 5 achieves maximum efficiency since CRATE, CRATE and SE are equal to 1. CHPP 2 achieves an efficiency factor of 1 only with variable returns on scale. With a constant return of scale, the efficiency coefficient is 0.978; SE for CHP 2 is also 0.978. In this case, we can say that the sources of inefficiency for this enterprise are caused by unfavorable conditions. These sources are not caused by the inefficient operation of the enterprise, since the technical efficiency for this enterprise is equal to 1.

**Table 7.** Comparative table of performance indicators for input-oriented CCR and BCC models with two input and one output parameters.

| CHPP | CRSTE | VRSTE | SE | Scale Type |
|------|-------|-------|------|-----------|
| 1 | 0.873 | 0.893 | 0.978 | drs |
| 2 | 0.978 | 1.000 | 0.978 | drs |
| 3 | 0.466 | 0.761 | 0.612 | irs |
| 4 | 0.872 | 0.891 | 0.978 | drs |
| 5 | 1.000 | 1.000 | 1.000 | - |
| 6 | 0.442 | 0.761 | 0.581 | irs |

CRSTE—constant return of scale technical efficiency, VRSTE—variable return of scale technical efficiency, SE—scale efficiencies, DRS—decreasing return of scale, IRS—increasing return to scale, CRSTE—technical efficiency from CRS DEA, VRSTE—technical efficiency from VRS DEA, Scale—scale efficiency = CRSTE/VRSTE.

Moreover, in the table, we can see a comparison of the performance indicators CRSTE, VRSTE, and SE for the remaining CHP, and we can also determine the type of return on scale: decreasing return to scale or increasing return to scale.

For example, CHPP 1 has the following performance indicators: CRSTE = 0.873, VRSTE = 0.893, and SE = 0.978. Analyzing these performance indicators, we can state the following: The sources of inefficiency for this enterprise are caused more by the inefficient operation of the enterprise than by unfavorable conditions, since VRSTE = 0.893, which is significantly less than 1. The indicator SE = 0.978, which is almost equal to 1. The most inefficient enterprises from the analyzed sample are CHPPs 3 and 6. Moreover, as can be seen from Table 7, their inefficiency is caused by both the inefficient operation of the enterprise and unfavorable conditions.

Analyzing the type of return of scale for this sample, we can say that CHPPs 3 and 6 are at the point of return to scale. The same CHPPs as described above have slack movements. CHPPs 1, 2, and 4 are at the point of decreasing return to scale.

### 5.4. Experiment 4—The Output-Oriented BCC Model with Two Input and One Output Parameters

We now build a basic output-oriented BCC model. The goal for this experiment will be to increase the supply of thermal energy to heat networks.

We calculate the efficiency using the output-oriented BCC model with two input and one output parameters.

This table shows that, according to this model, two enterprises are also effective: CHPPs 2 and 5, having an efficiency score of 1. They do not require setting an output score. Therefore, in the table, the output indicator (y1) remains unchanged for CHPPs 2 and 5. For all other CHPPs, the (y1) indicator changes. Table 8 shows the (y1) values required to achieve the efficiency of the CHPP data. For example, the efficiency indicator calculated using the DEA method for CHPP 1 is 0.925 with the maximum possible being 1. The output indicator (y1) is initially equal to 3635.

**Table 8.** Efficiency calculation based on the output-oriented BCC model with two input and one output parameters.

| № CHPP | Eff | O (y1) | I (x1) | I (x2) |
|---|---|---|---|---|
| 1 | 0.925 | 3931.163 | 1472.188 | 642.000 |
| 2 | 1.000 | 4519.000 | 1530.000 | 790.000 |
| 3 | 0.546 | 1974.169 | 752.000 | 394.641 |
| 4 | 0.923 | 3927.191 | 1554.000 | 641.000 |
| 5 | 1.000 | 4380.000 | 1405.000 | 767.000 |
| 6 | 0.514 | 1949.111 | 752.000 | 386.000 |

In order to increase the efficiency of this CHPP 1 to 1, and so make the enterprise fall on the efficiency frontier built on this sample, it is necessary to increase output (y1) to 3931.163. In this case, the efficiency coefficient calculated using the DEA method will be equal to 1, and, accordingly, the enterprise will become efficient.

The input (x1) indicator for the BCC model also changed from 1554 to 1472.188 as, according to the input indicator (x1), there was a slack movement equal to −81.813. The input indicator (x2) remains unchanged.

Table 9 presents the slack movement for the indicators of the studied sample.

**Table 9.** Slack movement for the indicators of the output-oriented BCC model with two input and one output parameters.

| № CHPP | O (y1) | I (x1) | I (x2) |
|---|---|---|---|
| 1 | - | −81.813 | - |
| 2 | - | - | - |
| 3 | - | - | −6.359 |
| 4 | - | −82.203 | - |
| 5 | - | - | - |

Table 9 shows slack movement is present for CHPPs 1, 3, and 4. The data for CHPPs 1 and 4 tend towards the indicators of CHPP 26. CHPP 3 tends towards the indicators of CHPP 27. Thus, CHPPs 1 and 4 tend to the nearest effective frontier point, which is CHPP 26 for them. CHPP 3 tends to the nearest effective frontier point, which for it is CHPP 26. Accordingly, this slack movement appears.

Below we present a table comparing performance indicators for the output-oriented CCR and BCC models with two input and one output parameters.

As the table shows, for this sample CHPP 5 is an effective enterprise according to the presented performance indicators: CRSTE, VRSTE, and SE. It achieves maximum efficiency in its work. CHPP 2 achieves an efficiency factor of 1 only with variable returns on scale. With a constant return on scale, the efficiency coefficient is 0.978. SE for CHPP 2 is also 0.978. In this case, we can say that the sources of inefficiency for this enterprise are caused by unfavorable conditions, and are not caused by the inefficient operation of the enterprise, since the technical efficiency for this enterprise is equal to 1.

All performance indicators for the remaining CHPPs are below 1 and are also presented in Table 10. For example, CHPP 1 has the following performance indicators: CRSTE = 0.873 and VRSTE = 0.925, SE = 0.945. Analyzing these performance indicators, we can say the following: The sources of inefficiency for this enterprise are caused more by the inefficient operation of the enterprise than by unfavorable conditions. The most inefficient enterprises from the sample analyzed, as in the previous experiment, are CHPPs 3 and 6. Moreover, as can be seen from Table 7, their inefficiency is caused by both the inefficient operation of the enterprise and unfavorable conditions.

**Table 10.** Comparative table of performance indicators for output-oriented CCR and BCC models with two input and one output parameters.

| CHPP | CRSTE | VRSTE | SE | Scale Type |
|:---:|:---:|:---:|:---:|:---:|
| 1 | 0.873 | 0.925 | 0.945 | DRS |
| 2 | 0.978 | 1.000 | 0.978 | DRS |
| 3 | 0.466 | 0.546 | 0.854 | IRS |
| 4 | 0.872 | 0.923 | 0.944 | DRS |
| 5 | 1.000 | 1.000 | 1.000 | - |
| 6 | 0.442 | 0.514 | 0.861 | IRS |

In the previous experiment, it was observed that CHPPs 1, 2, and 4 were at the point of decreasing scale effect (decreasing return to scale). CHPPs 3 and 6 were at the point of increasing scale effect (increasing return to scale).

*5.5. Experiment 5—The Output-Oriented BCC Model with Two Input and Two Output Parameters*

We conducted experiments on the basic CCR models with an input and output orientation with two inputs and one output. Conclusions were drawn from these experiments about the objects of the studied sample. There was also a comparison of performance indicators, and we calculated indicators of inputs and outputs when using different DEA models and their orientations.

Depending on the purpose of the DM, inputs, and outputs, models and their orientation may vary. It was previously mentioned that the environmental component is important when evaluating the efficiency of the CHP. Therefore, we believe that it is advisable to add another output, depending on the purpose of the decision maker, that is, whether it is important in these conditions to regulate the mass of emissions into the atmosphere. Then, it is also possible to use a model with two inputs and two outputs, adding as an output an indicator of accounting for emissions into the environment. Namely, as input indicators, we also use the following:

(1)  The available thermal power of the equipment (Gcal/hour)—input ($x1$).
(2)  The consumption of conventional fuel per released fuel cell (thousand tons/year)—input ($x2$).

As output indicators we use the following.

(1)  Supply of thermal energy to heat networks (thousand Gcal)—output ($y1$).
(2)  The mass of the emission (thousand tons per year)—output ($y2$).

We now build a basic output-oriented BCC model. The goal for this experiment will be to increase the supply of thermal energy to heat networks and reduce the mass of the emission.

We calculate the efficiency using the output-oriented BCC model with two input and two output parameters.

We do not give the full results of the CCR model experiments separately but present only a comparative table of performance indicators for the CCR and BCC models. In addition, as we have seen from previous experiments, our sample contains CHPPs that have decreasing and increasing scale effects. Accordingly, it is more expedient to measure efficiency using the BCC model for this sample.

In Table 11, the O ($y2$) column shows the values of the outputs ($y2$) emission mass (thousand tons per year) for each CHP. In accordance with the selected orientation of the model, the output indicators will increase. However, according to the goal set, the emission mass index needs to be reduced. As described above, in such cases, the deviation indicator from the maximum value of this indicator is used for the entire sample. In our case, the maximum value is 17.4, which, rounded up to an integer, will be 18. Next, we present

rounded indicators of deviation from the maximum indicator for each CHPP. The data is presented in Table 12.

**Table 11.** Initial data on two input indicators and two output indicators of the studied CHPP sample.

| № CHPP | O (y1) | O (y2) | I (x1) | I (x2) |
|--------|--------|--------|--------|--------|
| 1 | 3635 | 14.9 | 1554 | 642 |
| 2 | 4519 | 13.5 | 1530 | 790 |
| 3 | 1077 | 7.30 | 752 | 401 |
| 4 | 3626 | 15.2 | 1554 | 641 |
| 5 | 4380 | 13.9 | 1405 | 767 |
| 6 | 1001 | 7.90 | 752 | 386 |

**Table 12.** Initial data (deviation coefficients) for the output (y2) of the studied CHPP sample.

| CHPP 1 (y2) | CHPP 2 (y2) | CHPP 3 (y2) | CHPP 4 (y2) | CHPP 5 (y2) | CHPP 6 (y2) |
|-------------|-------------|-------------|-------------|-------------|-------------|
| 3 | 5 | 11 | 3 | 4 | 10 |

The indicators presented in Table 12 will be used to calculate the effectiveness of the BCC model.

We present the results of efficiency calculations based on the BCC model in Table 13.

**Table 13.** Efficiency calculation based on the output-oriented BCC model with two input and two output parameters.

| № CHPP | VRSTE | y1 | y2 | x1 | x2 |
|--------|-------|------|------|------|------|
| 1 | 0.925 | 3931.163 | 4.075 | 1472.188 | 642 |
| 2 | 1.000 | 4519.000 | 5.000 | 1530.000 | 790 |
| 3 | 1.000 | 1180.000 | 11.000 | 752.000 | 355 |
| 4 | 0.923 | 3927.191 | 4.069 | 1471.797 | 641 |
| 5 | 1.000 | 4380.000 | 4.000 | 1405.000 | 767 |
| 6 | 0.909 | 1180.000 | 11.000 | 752.000 | 355 |

According to this model, three enterprises are effective: CHPPs 2, 3, and 5. They have an efficiency indicator of 1. They do not require setting output indicators. Therefore, in the table the outputs indicators (y1) and (y2) remain unchanged. For all other CHPPs, the indicator (y1) and (y2) changes.

In Table 13, the values of outputs (y2) should be explained. The output (y2) values for CHPPs 2, 3, and 5 have not changed. Accordingly, the outputs (y2) change for CHPPs 1, 4, and 6. The changes can be explained. For CHPP 1 it is required to reduce (y2) by 1.075; for CHPP 4 it is required to reduce (y2) by 1.069; and for CHPP 6 it is required to reduce (y2) by 1. At the same time, the outputs (y1) for these CHPPs will also change to increase the efficiency indicator to 1. We now explain the changes in outputs (y1). For CHP 1 it is required to increase y2 by 296.163; for CHPP 4 it is required to increase (y2) by 301.191; and for CHPP 6 it is required to increase (y2) by 179.

The input parameters (x1) and (x2) according to the BCC model have also changed for some CHPPs due to slack movement. For CHPPs 2 and 5, slack movement is not observed in any indicator. For the rest of the CHPPs, such changes are present. For CHPP 1, as a result of adjusting indicators to achieve an efficiency coefficient of 1, slack movement occurs with output (y2) and input (x1). Output (y2) decreases by 0.831, and input (x1) decreases by 81,812. For CHPP 3, as a result of adjusting indicators to achieve an efficiency coefficient of 1, slack movement occurs with output (y1) and input (x2). Output (y1) increases by 103

and input (x2) decreases by 46. For CHPP 4, as a result of adjusting indicators to achieve an efficiency coefficient of 1, slack movement occurs with output (y2) and input (x1). Output (y2) decreases by 0.82, and input (x2) decreases by 82.203. For CHPP 6, as a result of adjusting indicators to achieve an efficiency coefficient of 1, slack movement occurs with output (y1) and input (x2). Output (y1) increases by 78.9 and input (x2) decreases by 31.

As described above, the BCC model helps to calculate efficiency considering VRS. The CCR model calculates efficiency using CRS. Below, we present a comparative table of performance indicators for the output-oriented CCR and BCC models with two input and one output parameters.

As we can see from the table, for this sample, CHPPs 2, 3, and 5 are effective enterprises according to the performance indicators of CRSTE, VRSTE, and SE. That is, they achieve maximum efficiency in their work. CHPP 6 achieves an efficiency factor of 1 only at SE. This means that for this enterprise, unfavorable conditions have no effect on its efficiency. With a constant return on scale and a variable return on scale, the efficiency coefficients are the same and equal to 0.909. Thus, the inefficiency of this enterprise is wholly caused by the inefficient operation of the enterprise. All performance indicators for the remaining CHPPs are below 1 and are also presented in Table 14. Nonetheless, as we can see from the table, the performance indicators for all the enterprises in the sample are considerably high.

**Table 14.** Comparative table of performance indicators for output-oriented CCR and BCC models with two input and two output parameters.

| CHPP | CRSTE | VRSTE | SE | Scale Type |
|------|-------|-------|-------|-----------|
| 1 | 0.892 | 0.925 | 0.965 | DRS |
| 2 | 1.000 | 1.000 | 1.000 | - |
| 3 | 1.000 | 1.000 | 1.000 | - |
| 4 | 0.891 | 0.923 | 0.965 | DRS |
| 5 | 1.000 | 1.000 | 1.000 | - |
| 6 | 0.909 | 0.909 | 1.000 | - |

In this experiment, we observe changes in the display of the scale effect. From Table 14, it can be seen that only CHPPs 1 and 4 are at the point of decreasing scale effect (decreasing return to scale).

In this experiment, it can be seen that the efficiency coefficients are significantly higher for the same sample as in previous experiments. There are also significantly more efficient enterprises. In comparison, with the previous experiment, where CHPPs 3 and 6 had a rather low efficiency of CRSTE and VRSTE (below 0.6), CHPP 3 has now become an efficient enterprise, and CHP P6 has CRSTE and VRSTE efficiency indicators of more than 0.9. The introduction of another output, which the model focuses on, yields a significant change in the indicators for the sample. Depending on the purpose of the decisionmaker, the input and output indicators are adjusted and the efficiency coefficients are changed according to the DEA model.

*5.6. Experiment 6—The Input -Oriented BCC Model with Four Input Indicators and One Output Indicator*

We conducted experiments on the basic models of BCC and CCR with an input and output orientation, using two inputs and one output and two inputs and two outputs, respectively. From these experiments, conclusions were drawn about the objects of the sample studied and changes in performance indicators.

Changing inputs and outputs can occur when the target of the decision maker changes. It is expedient to reflect certain inputs and outputs, when solving a certain task of increasing efficiency. In the study of the CHPP of the heating system, it is possible to replace the input of the available thermal power of the equipment with three inputs: the heating capacity

of turbo generators (Gcal/hour); the power of peak hot water boilers (Gcal/hour); and the power of steam boilers (Gcal/hour). This will give us the opportunity to adjust the efficiency of work for each type of equipment. Depending on the purpose of the decision maker, you can also use a model with four inputs and one output. Namely, as input indicators we use the following:

(1)    Heating capacity of turbo generators (Gcal/hour)—input (x1).
(2)    Power of peak hot water boilers (Gcal/hour)—input (x2).
(3)    Power of steam boilers (Gcal/hour)—input (x3).
(4)    Consumption of conventional fuel for the released fuel (thousand tons/year)—input (x4).

As an output indicator, we use the following:

(1)    Heat supply to the grid (thousand Gcal)—output (y1).

This combination of inputs and outputs will help adjust the efficiency of the equipment and adjust fuel consumption and heat release when achieving maximum efficiency of the CHPP.

The initial data are presented in Table 15.

**Table 15.** Initial data on four input indicators and one output indicator of the CHPP sample.

| № CHPP | O (y1) | I (x1) | I (x2) | I (x3) | I (x4) |
|---|---|---|---|---|---|
| 1 | 3635 | 850 | 210 | 230 | 642 |
| 2 | 4519 | 909 | 226 | 270 | 790 |
| 3 | 1077 | 270 | 440 | 42 | 401 |
| 4 | 3626 | 850 | 210 | 230 | 641 |
| 5 | 4380 | 909 | 226 | 270 | 767 |
| 6 | 1001 | 270 | 440 | 42 | 386 |

We now build a basic input-oriented BCC model. The goal for this experiment will be to reduce the power of equipment and fuel consumption while maintaining the indicator of heat supply to the grid. We will calculate the efficiency using the input-oriented BCC model with four input indicators and one output indicator of the studied CHPP sample.

We present the results obtained in Table 16.

**Table 16.** Efficiency calculation based on the input-oriented BCC model with four input indicators and one output indicator.

| № CHPP | O (y1) | I (x1) | I (x2) | I (x3) | I (x4) |
|---|---|---|---|---|---|
| 1 | 3635 | 850.000 | 210.000 | 230.0 | 595.681 |
| 2 | 4519 | 909.000 | 226.000 | 270.0 | 790.000 |
| 3 | 1077 | 190.000 | 340.000 | 42.0 | 283.902 |
| 4 | 3626 | 850.000 | 210.000 | 230.0 | 593.122 |
| 5 | 4380 | 899.769 | 223.705 | 263.9 | 759.211 |
| 6 | 1001 | 190.000 | 340.000 | 42.0 | 281.271 |

We also present a table of performance indicators for the input-oriented CCR and BCC models with four input and one output parameters, providing a visual opportunity to see changes in input indicators depending on performance indicators.

As can be seen from Table 17, according to the BCC VRSTE model, five enterprises are effective: CHPPs 1, 2, 3, 4, and 6. They have an efficiency score of 1. However, for four of them (CHPPs 1, 3, 4, and 6) there is slack movement. Only for CHPP 2 do the input indicators not change, with slack movement also not being observed for any indicator. For the rest of the CHPPs, such changes are present.

**Table 17.** Comparative table of performance indicators for input-oriented CCR and BCC models with four input indicators and one output parameter.

| CHPP | CRSTE | VRSTE | SE | Scale Type |
|------|-------|-------|-------|-----------|
| 1 | 0.957 | 1.00 | 0.957 | irs |
| 2 | 1.000 | 1.00 | 1.000 | - |
| 3 | 0.627 | 1.00 | 0.627 | irs |
| 4 | 0.956 | 1.00 | 0.956 | irs |
| 5 | 0.989 | 0.99 | 0.999 | irs |
| 6 | 0.582 | 1.00 | 0.582 | irs |

For CHP 1, as a result of adjusting indicators to achieve an efficiency coefficient of 1, slack movement occurs with input (x1) and is 46.319. For CHPP 3, as a result of slack movement, input (x1) decreases by 80, input (x2) decreases by 100, and input (x4) decreases by 117.098. For CHPP 4, as a result of adjusting indicators to achieve an efficiency coefficient of 1, slack movement occurs with input (x1) and is 46.319. For CHPP 5, as a result of slack movement, input (x3) decreases by 3.341. Finally, for CHPP 6, as a result of adjusting indicators to achieve an efficiency factor of 1, slack movements occur with input (x1), which decreases by 80; input (x2) decreases by 100 and input (x4) decreases by 104,729.

As can be seen from Table 16, the input (x1–x4) indicators for CHPPs 1, 2, 3, 4, and 6 have not changed. As described above, only slack movements occurred there. For CHPP 5, the input indicators (x1–x4) change. Input (x1) decreases by 9.231, input (x2) decreases by 2.295, input (x3) decreases by 2.742, and input (x4) decreases by 7.789.

As we can see from Table 17, for this sample, only CHPP 2 is an effective enterprise. According to the presented performance indicators of CRSTE, VRSTE, and SE, CHPP 2 achieves maximum efficiency in operation. All performance indicators for the remaining CHPPs for CRSTE and SE are below 1 and are also presented in the table. Table 17 shows CHPPs 1, 3, 4, 5, and 6 are at the point of increasing return to scale.

As can be seen from Table 17, the performance indicators for CHPPs 1, 3, 4, and 6 are equal to 1, but the performance indicators of CRSTE and SE are below one. This suggests that these enterprises are working quite efficiently. Hence, the inefficiency of these enterprises is wholly caused by unfavorable conditions. These have a particularly strong effect on CHPPs 3 and 6, since their SE is 0.627 and 0.582, respectively.

In this experiment, it can be seen that the efficiency coefficients for VRSTE are significantly higher for the same sample as in previous experiments, being 1 for five CHPPs. However, only one CHP 2 achieves maximum efficiency in operation. If in previous experiments CHPP 5 was effective and achieved maximum efficiency in operation, then, in this experiment, it is the only CHPP that has a VRSTE efficiency coefficient below 1. As can be seen from Table 17, in this experiment five CHPPs are at the point of increasing scale effect (increasing return to scale), while, in experiments 3 and 4 only CHPPs 3 and 5 were at the point of increasing return to scale.

## 6. Discussion

In this article, we investigated the theoretical aspects of evaluating the effectiveness of complex systems, as well as conducting an analysis of literary sources on the subject of the study. The concept of efficiency and productivity of complex systems based on the DEA method was analyzed. The DEA method itself is described in sufficient detail. The basic DEA models are described and their comparative analysis is presented. The methodology of applying the DEA method in a specific field was presented. We also demonstrated the application of this methodology in the fuel and energy complex, a context in which the DEA method had not previously been used. The application of the methodology for assessing complex technical systems was presented in the district heating system using the example of the CHPP. A total of 27 CHPPs were represented in the study sample,

for which experiments were conducted to improve the efficiency of the CHPP with the setting of different DM goals. As a result, when the goal was changed, different DEA models were built. Calculations of efficiency indicators were carried out when changing inputs and outputs, according to the orientation of the model for the same studied sample. The indicators of inputs and outputs were calculated to achieve maximum efficiency—an efficiency indicator equal to 1. The CCR and BCC models were tested, with an input orientation and an output orientation. The models were built according to variations of inputs and outputs, depending on the purpose of the DM. Experiments were carried out on combinations of two inputs and one output, two inputs and two outputs, and four inputs and one output, respectively. The following performance indicators were calculated in the experiments: technical efficiency from CRS DEA, technical efficiency from VRS DEA, and scale efficiencies. It was also determined at which point of the scale effect (decreasing return to scale/increasing return to scale) each of the enterprises in the studied sample is located.

The article presents 6 experiments. The first four experiments were carried out on the sample analyzed with two inputs (the available thermal power of the equipment and the consumption of conventional fuel for the released fuel cell) and one output (the release of thermal energy to heat networks). In these experiments, efficiency calculations were carried out using the CCR and BCC models. We compared the results of efficiency calculations using these models. We compared the inputs and outputs of inefficient DMUs when they achieved efficiency. We analyzed the reasons for inefficiency in different experiments. In the experiments of the paper we detailed the results using these models. The limitations of using these models are described in the paper.

When setting the goal of reducing the consumption of conventional fuel for the released fuel cell and reducing the available thermal power of the equipment, the following results were obtained. The input-oriented DEA model was set. Under experiments 1 and 3, CHPP 5 is an effective enterprise according to the presented performance indicators: CRSTE, VRSTE, and SE. That is, it achieves maximum efficiency in its work. CHPP 2 achieves an efficiency factor of 1 only with VRS. With a CRS, the efficiency coefficient is 0.978. SE for CHPP 2 is also 0.978. In this case, we can say that the sources of inefficiency for this enterprise are caused by unfavorable conditions. However, their influence is low, since the efficiency coefficient is quite high and is equal to 0.978. This enterprise works reasonably efficiently, since the technical efficiency for this enterprise is also equal to 1. The most inefficient enterprises from the sample are CHPPs 3 and 6. Moreover, their inefficiency is caused by both the inefficient operation of the enterprise and unfavorable conditions. For all inefficient enterprises, the required input indicators were calculated to achieve an efficiency coefficient equal to 1. Analyzing the type of return on scale according to these experiments, we can say that CHPPs 3 and 6 are at the point of increasing return to scale, while CHPPs 1, 2, and 4 are at the point of decreasing return to scale. Slacks movements are observed for CHPP 3 and 6 for the BCC model. When calculating using the CCR model, slacks movements are not observed.

In experiments 2 and 4, the goal was to increase the supply of thermal energy to heat networks. To this end, efficiency calculations were carried out and the following results were obtained. An output-oriented DEA model was set. Under experiments 2 and 4, CHPP 5 is an effective enterprise according to the presented performance indicators: CRSTE, VRSTE, and SE. The same results were obtained for experiments 1 and 3. Identical efficiency coefficients were also obtained for CHPP 2. For the rest of the enterprises in the sample, the CRSTE performance indicators are also identical, while the performance indicators of VRSTE and SE differ. The most inefficient enterprises from the sample, as in the previous experiment, are CHPPs 3 and 6. Their inefficiency is also caused by both inefficient operation of the enterprise and unfavorable conditions. For all inefficient enterprises, the required output indicators were calculated to achieve an efficiency coefficient equal to 1.

In the experiment, it is observed that CHPPs 1, 2, and 4 are at the point of decreasing return to scale. CHPP 3 and 6 are at the point of increasing return to scale. Slack movements

were already observed for CHPPs 1, 3, and 4 for the BCC model. For experiments 1 and 3, only CHPPs 3 and 6 presented slack movements, while no slack movements are found in calculations using the CCR model.

In experiment 5, another output indicator was added. This was done in order to see the impact of the environmental factor on the efficiency of the CHPP, adjusting the performance of the CHPP plant while reducing emissions into the atmosphere. In this experiment, the goal was to increase the supply of thermal energy to heat networks and reduce the mass of harmful substances released into the atmosphere. In accordance with the goal set, an output-oriented model with two inputs and two outputs was used. As input indicators, the following were used: the available thermal power of the equipment, the consumption of conventional fuel for the released fuel cell. As output indicators, the following were used: the release of thermal energy to heat networks and the emission mass. The efficiency calculations using the DEA method for CCR and BCC models yielded the following data. In this experiment, with the same sample, three enterprises are already effective enterprises according to the performance indicators of CRSTE, VRSTE, and SE, namely CHPPs 2, 3, and 5. They achieve maximum efficiency in their work. CHPP 6 achieves an efficiency factor of 1 only at SE. This means that for this enterprise, unfavorable conditions have no effect on its efficiency. With a constant return on scale and a variable return on scale, the efficiency coefficients are the same and equal to 0.909. Thus, the inefficiency of this enterprise is wholly caused by the inefficient operation of the enterprise. All performance indicators for the remaining CHPPs are below 1. For these, the required output indicators were calculated to achieve an efficiency coefficient equal to 1.

In this experiment, we observe changes in the display of the scale effect. Now, only CHPPs 1 and 4 are at the point of decreasing scale effect (decreasing return to scale). In addition, for CHPPs 2 and 5 slack movement is not observed on any indicator. For the rest of the CHPPs such changes are present.

In this experiment, it can be seen that the efficiency coefficients are significantly higher for the same sample as in previous experiments. Additionally, there are significantly more efficient enterprises. In comparison with the previous experiment, where CHPPs 3 and 6 had a rather low efficiency of CRSTE and VRSTE (below 0.6), CHPP 3 is now an efficient enterprise. CHPP 6 has CRSTE and VRSTE performance indicators of more than 0.9. Accordingly, the introduction of another output, which the model focuses on, yielded a significant change in the performance indicators for the sample. In this case, with the introduction of the emission mass output, there was a significant increase in efficiency indicators. This may indicate that enterprises in the field of environmental factors are working quite efficiently. However, in the absence of this output, the same enterprises have lower performance indicators. All these features should be considered when compiling the algorithm of the DSS, adding possible options for changing the goal when solving the problem of increasing efficiency by the DEA method.

In experiment 6, the input of the available thermal power of the equipment was replaced by three inputs: the heating power of turbo generators, the power of peak hot water boilers, and the power of steam boilers. This was done in order to be able to adjust the efficiency of work for each type of equipment used at the CHPP. In this experiment, the goal was to reduce the power of the equipment and the consumption of conventional fuel for the released fuel cell. Accordingly, the DEA model with four inputs and one output, oriented to the input, was used. As input indicators, the following were used: the heating capacity of turbo generators, the power of peak hot water boilers, the power of steam boilers, and the consumption of conventional fuel for the released fuel cell. As an output indicator, we used the heat energy released to the grid. This combination of inputs and outputs helped adjust the efficiency of each type of equipment, adjusting fuel consumption when achieving maximum efficiency of the CHPP. As a result of efficiency calculations using the DEA method for CCR and BCC models, the following data were obtained. In this experiment, with the same sample, only CHPP 2 is an effective enterprise according to the performance indicators of CRSTE, VRSTE, and SE. That is, CHPP 2 achieves maximum

efficiency in operation. In addition, in this experiment, 5 enterprises are effective according to VRSTE: CHPPs 1, 2, 3, 4, and 6, having an efficiency score of 1. However, for four of them (CHPPs 1, 3, 4, and 6), there is slack movement. Only for CHPP 2 do the input indicators not change, while slack movement is also not observed for any indicator. For the rest of the CHPPs, such changes are present.

The efficiency indicators for CHPP 1, 3, 4, and 6 are equal to 1. However, the efficiency indicators of CRSTE and SE are below 1. This suggests that these enterprises are working quite efficiently. The inefficiency of these enterprises is wholly caused by unfavorable conditions. They have a particularly strong effect on CHPP 3 and CHPP 6, since their SE is 0.627 and 0.582, respectively.

In this experiment, it can be seen that the efficiency coefficients for VRSTE are significantly higher for the same sample as in previous experiments. For five CHPPs, these are 1, but only one CHPP, number 2, achieves maximum efficiency in operation. If in previous experiments CHPP 5 was effective and achieved maximum efficiency in operation, then in this experiment it is the only CHPP that has a VRSTE efficiency coefficient below 1. In this experiment, five CHPPs are at the point of increasing scale effect (increasing return to scale). Meanwhile, in experiments 3 and 4 only CHPPs 3 and 5 were at the point of increasing scale effect (increasing return to scale).

Now, let us compare the data from experiment 3 and this experiment. In experiment 3 the goal was the same, but only in this experiment, the input indicator of the available thermal power of the equipment was divided into three indicators: the heating capacity of turbo generators, the power of peak hot water boilers, and the power of steam boilers. This was done to make it possible to regulate the power of each type of equipment at the CHPP. When comparing the performance indicators according to these experiments, it is clear there is a significant difference, both in terms of the scale effect and in the efficiency of the enterprises in the same sample. For example, the most efficient enterprise in experiment 3 was CHPP 5, while, in this experiment, it is CHPP 2. The VRSTE efficiency is equal to 1 for CHPPs 1, 3, 4, 5, and 6 in this experiment. In experiment 3, the VRSTE efficiency was equal to 1 only for CHPPs 2 and 5.

The inefficiency of enterprises in this experiment is mainly caused by unfavorable conditions. The inefficiency of enterprises in experiment 3 is mainly caused by both unfavorable conditions and inefficient operation of enterprises. In addition, in this experiment, all enterprises in the sample have an increasing return to scale. In contrast, in experiment 3, only CHPPs 3 and 6 have an increasing return to scale. The remaining CHPPs are at the point of decreasing return to scale.

The limitations of the algorithm were not revealed. When using this algorithm, you should accurately understand the purpose of the decision maker and at the expense of which tools of the DEA model the goal can be achieved. In the experimental part, we examined in detail the various options for goals and how they can be solved using the DEA method. It is also necessary to know how the investigated DMUs function in order to correctly select the inputs and outputs.

The algorithm can be applied to other technical systems.

Readers do not need additional parameters to conduct such experiments. Data sources are needed—data on the inputs and outputs of the DMU. No more data are required. Readers can repeat the experiments described in the article without problems, since they are analyzed in sufficient detail.

Using the DEA method and conducting experiments makes it possible to program a decision support system. This will make it possible to apply this algorithm to enterprises in the form of a decision support system and increase the adaptability and manageability of enterprises of the fuel and energy complex.

### 7. Conclusions and Future Work

In this article, we have described in sufficient detail the algorithm of using the DEA method for district heating enterprises. Therefore, readers can easily use the results of these experiments and repeat such calculations independently.

Our experiments have demonstrated how much the performance indicators of the CHPP can change. In accordance with these indicators, the settings of input and output indicators for the sample are also changed to achieve efficiency. For example, for one experiment, the same CHPP can achieve maximum efficiency in operation, and for another it can be ineffective. When the goal to be achieved changes the conditions for the DEA task change. The results of calculations of the effectiveness of DEA models for the same sample may change dramatically.

It can be concluded that for the possibility of application, as well as for the correctness of the application of the DEA method, an understanding of the application of this method to various models in a particular field is required. As well as understanding how and when the goal changes, it is necessary to understand that the indicators of inputs and outputs, the model itself, and the orientation of the DEA model change. The district heating system is a rather complex system with a large number of technical indicators. Therefore, achieving the efficiency of such systems is a complex and time-consuming process. The introduction of new methods for evaluating the effectiveness of such systems will solve a number of problems. In the future, it will lead to automation of management processes, due to the possibility of creating a DSS to increase efficiency based on the DEA method. It will also lead to an increase in the adaptability of the system to external factors due to the possibility of changing the settings of the DEA method when changing the goals of the DM.

The data obtained from the experiments made it possible to observe changes in performance indicators when changing the goal of DM in the same sample. It was clearly demonstrated that the use of various DEA models and their orientations leads to changes in the settings of input and output indicators. Thus, only with a full understanding of the patterns and performance indicators of the studied field is the correct use of DEA models possible. The objectives of the DM may change in the course of the functioning of the enterprises due to rapidly changing external factors in the operating environment. Therefore, in these conditions, it is important to change the settings for building DEA models for specific enterprises, as well as their input and output parameters. The methodology of using DEA models for evaluating efficiency presented in this article makes it possible to apply the DEA method to improve the efficiency of fuel and energy companies, a context in which this method and methodology have not previously been used.

In the article, we have shown how, when the goal of the decision maker changes, the performance indicator can change. This is a very important conclusion for understanding the change in efficiency when the operating conditions of the system change. Now, it has become clear how we could adjust the effectiveness when changing the goal of the decision maker. The results obtained in the experimental part will be the basis for programming an automated decision-making system in the future.

There are 2 aspects on which we can continue working:

1. To develop an automated decision support system for fuel and energy complex enterprises based on the algorithm and experimental results.

2. When using the DEA method, there are several usage restrictions. Further work can be continued on the development of a new efficiency method that can minimize the limitations of its use.

The algorithm presented, the data obtained and the conclusions from the presented experiments will help in using the DEA method to increase the efficiency of fuel and energy companies. In the continuation of this study, they will form the basis of the DSS algorithm for combined heat and power plants. In further research, an automated DSS will be developed based on the results obtained. This DSS will be based on an algorithm for applying different DEA models depending on the performance of the CHP. It will also be possible to integrate the DSS based on the DEA method into the management system of

modern CHPPs. This DSS will help automatically calculate and adjust the desired change in the performance of the system elements from changes in the indicators of inputs and outputs over time and depending on changes in external conditions. This will automate the process of increasing the efficiency of fuel and energy companies based on the DEA method.

**Author Contributions:** M.P., A.S. and I.M.-B. were engaged in the formulation of research goals and objectives, conducted a simulation of the algorithm for using DEA models to calculate the efficiency of fuel and energy complex enterprises, organized experiments, and prepared a publication. S.E. organized the resource provision of information and reference materials for research and was engaged in the preparation and editing of the publication. R.K. and R.P. were responsible for statistics, were engaged in data storage and processing, and carried out data analysis, processing, and visualization. They engaged in the design of the publication. All authors have read and agreed to the published version of the manuscript.

**Funding:** This research was founded by MCIN/AEI/10.13039/501100011033/ and ERDF A way to do Europe grant number AwESOMe Project PID2021-122215NB-C33 and in part by the Ministry of Science and Higher Education of the Russian Federation grant number 075-15-2022-1121. The APC was funded by MCIN/AEI/10.13039/501100011033/ and ERDF A way to do Europe grant number AwESOMe Project PID2021-122215NB-C33.

**Data Availability Statement:** The data used in the paper are publicly available at the following link: Administration of the City of Krasnoyarsk. The Project "Heat Supply Scheme of the City of Krasnoyarsk Until": [Electronic Resource]. 2015. Available online: http://www.admkrsk.ru/citytoday/municipal/energy/teplosn/Documents/%D0%9F%D1%80%D0%BE%D0%B5%D0%BA%D1%82%20%D1%81%D1%85%D0%B5%D0%BC%D1%8B%20%D1%82%D0%B5%D0%BF%D0%BB%D0%BE%D1%81%D0%BD%D0%B0%D0%B1%D0%B6%D0%B5%D0%BD%D0%B8%D1%8F.pdf (accessed on 12 September 2023).

**Acknowledgments:** This work was supported by the Ministry of Science and Higher Education of the Russian Federation (Grant № 075-15-2022-1121) and in part by the Spanish Ministry of Science and Innovation and the European Regional Development (ERDF) Funds (Grant PID2021-122215NB-C33).

**Conflicts of Interest:** The authors declare no conflict of interest.

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
