# Peer review of "Algorithm for Application of a Basic Model for the Data Envelopment Analysis Method in Technical Systems"

_algorithms, doi:10.3390/a16100460_

Round 1
Reviewer 1 Report
This paper proposes to use Data Envelopment Analysis (DEA) to define an algorithm to solve the problem of efficiency of high fuel energy complex enterprises, considering all the technical indicators of such enterprises to optimize their work. Especially taking cogeneration as an example, the algorithm is first applied to the cogeneration heating system, and the experimental research is carried out. According to the case study, the efficiency index is calculated, and the input and output of the model are changed to achieve the maximum efficiency of the system. The final models have been tested with good results. The proposed method and experimental results make it possible for the first time to apply DEA methods to improve the efficiency of fuel and energy companies.
General comments
I was a bit disappointed, though, with the treatment of the development of the proposed method. Although the problem studied in this paper is novel, I think there is still room for improvement.
1、 The key words of the paper should be representative, accurate and have a moderate number. The number of keywords in this paper is too large, and there are duplication and other problems, such as Data Envelopment Analysis and DEA method, the author should re-select.
2、 In the introduction, the author should explain clearly the gap between the research in this paper and the existing research, and highlight the contribution of the paper, but in this part of the paper, there are almost no such statements.
3、 The second part of the paper is Basic concepts. In this part, the author should rethink the meaning of existence, because it is not his own innovation, whether it needs to be explained in detail or in a general way. As for the literature review before section 3.1 of the third part, can the author consider combining it with the second part and writing it in the form of literature review.
Another common problem worth mentioning is that the number of articles cited in the authors' literature reviews is mostly older articles. I suggest that the author read more literature in the last three years, which should give the author some new ideas and inspiration.
Finally, with regard to the editorial comments, it is suggested that the author adjust the aesthetics and clarity of the formulas and charts in the article. For example, the letter marks in Figure 2 overlap with lines, which is inconvenient for readers to read.
Good
Author Response
Please see the file that we attached.

Reviewer 2 Report
This article used DEA method to solve the problem of increasing the efficiency of enterprises of the fuel and energy complex. There are the following specific shortcomings.
(1) First of all, The introductory part of the article is too simply written, lacks logic and is rather empty in content. This is manifested in the following aspects:
① The entire Introduction section is short and does not provide a good introduction to the connection between the background of the study and the research methodology. Fewer references are cited and it is recommended to combine it with the third section 'Related work' to adjust the layout of the article.
② Lines 52-59 lack references. Please add relevant literature demonstrating that the effectiveness of planning in various activity fields is dependent on the operation of the system, thus better illustrating the need to improving the planning and management effectiveness in systems functioning.
③ The introduction to the DEA methodology should focus on the reasons why the methodology is applicable to the research context rather than the superiority of the methodology itself. Please adjust the length to fit the context of the article.
(2) .After line 114 ‘The management concept orientation of any type organization on the effectiveness of its activities requires the’, there is a missing element.
(3) The fourth and fifth parts of the article are structurally confusing, unframed, and a poor reading experience. Please adjust the paragraph structure and add subheadings to make the article structure clear
(4) There are many formatting problems in the References section, so please double check and correct each one. For example,
and,
In my opinion, This article has many flaws in details. Major review before publication.

Author Response
Please see the file that we attached.

Reviewer 3 Report
Based on the Manuscript “algorithms-2577665” provided, here are some questions for the authors to enhance the quality of the manuscript before re-submission
Research Gap and Motivation:
Could you elaborate on the specific challenges faced by enterprises in the fuel and energy complex that led you to choose the Data Envelopment Analysis (DEA) method for optimization?
What existing research or methods have been attempted to address the efficiency issues in these enterprises, and how does your proposed algorithm improve upon them?
Algorithm Clarity:
Can you provide a step-by-step breakdown of the proposed algorithm for applying DEA to technical systems? This will help readers better understand and replicate the process.
Are there any assumptions or limitations associated with the algorithm that need to be explicitly stated for a comprehensive understanding?
Experimental Design:
Could you provide more details about the conducted experiments on the combined heat and power plant? What were the specific inputs, outputs, and variables manipulated during the experiments?
Were there any particular criteria or benchmarks used to determine the "maximum efficiency of the system"? How were these criteria chosen?
Model Selection and Comparison:
The abstract mentions testing both the Charnes, Cooper, and Rhodes model and the Banker, Charnes, and Cooper model. Could you briefly explain the reasons behind selecting these models, and how they were applied in the context of your study?
How did you measure and compare the performance of these models? Did you encounter any challenges or limitations while implementing them?
Significance and Contribution:
How does the application of DEA to improve the efficiency of fuel and energy companies contribute to the broader field? What insights or practical implications does your methodology provide for decision-makers in these industries?
Replicability and Generalization:
Are there any specific technical details, parameters, or data sources that are necessary for readers to replicate your experiments in different settings?
How transferable is your proposed algorithm and methodology to other technical systems beyond the combined heat and power plant?
Discussion of Results:
In what ways do the "good results" obtained from testing the models contribute to the understanding of the enterprise's efficiency? Were there any unexpected findings that warrant further discussion?
Future Work:
Are there any aspects of your research that you believe require further investigation or refinement in future studies? For example, are there specific challenges or scenarios that the current algorithm doesn't address?
References:
Have you appropriately cited prior works related to both DEA and the fuel and energy complex to provide a comprehensive literature background?
Minor
Author Response
Please see the file that we attached.

Reviewer 4 Report
There are more observation for the paper proposal:
Page 3 (lines 113-114) : - missing end of the sentence: “The management concept orientation of any type organization on the effectiveness of its activities requires the” . Complete the sentence!
Page 4 (lines 184-197): -missing colon and numbering: Add a colon to the expression “they usually focus on 3 main aspects that must be considered when constructing 184 the DEA model [15]” and then insert in a list the three aspects with numbers or dashes (e.g. 1. What input and output indicators of the model will be used … 2. What determines the choice of a constant or variable scale effect …. 3. What determines the choice of the model orientation …).
Page 4 (lines 202): - missing point between 2 sentences: “… was named the CCR model The name of the model …” Insert point “.” between “model” and “The”
Page 5 (lines 230): - wrong parameter explanation: “?? is weighting of input factor i … “, “?? is weighting of output parameter …” No! In expression 2, in restriction and in expression 3 Ur is the weight for output and Vi the weight for input. Insert correct definitions!
Page 5 (lines 242-243): - duplicate sentence: “The founders of the model are Charnes, Cooper and Rhodes in 1978 [8]. In this regard, the model was named the CCR model. The name of the model is given as an abbreviation of the authors” you already said that (lines 201-203) remove duplicate sentences.
Page 6 (lines 303): - move title of section 2.3. to the next page.
Page 9 (lines 402): - one single member enumeration “Constant scale effect is applied in the models: CCR.” – in that case is not suitable to use “:” it is a single model… Just reformulate: “… is applied to the model CCR”.
Technical question: regarding the statements from page 12: “As output indicators, we use: 1) Supply of thermal energy to the grid (thousand Gcal). 2) The mass of the emission (thousand tons per year).” Why for a CHP you don’t have anything about electrical energy which CHP power plant provides? From my point of view this would be an important output indicator. Explain this omission.
Page 14 (lines 667-682): For me, this lines seems like a kind of general explanation about the implementation of algorithms. Honestly, the statement: "Be sure that DEA methods can be used in this field." on page 14 of a 30-page article that talks about the use of DEA for optimization at a power plant is mind-blowing for the reader... Isn't that what the article is about? I think that the article can be written in a more condensed form in which general explanations of this kind can be removed.
Throughout the article there are expressions suitable rather than an oral presentation of the type: “Move on to the experimental part.” , “Now move on to the scale effect.” “These will be the models: CCR and BCC.”, “For example, increase the number of CHPs.” “We can only use it in an output-oriented model. Where this indicator will remain unchanged.” “Now we move on to setting the goal of this study.” “The reason for inefficiency is the inefficient operation of the enterprise in the case of PTE. And the inefficiency is caused by unfavorable conditions in the case of SE.” etc . You must give up this oral expression in a written article and reformulate so that the expression appears continuous for a reader. For example: Instead of : “Move on to the experimental part.” you can say “Next, we will present the experimental part.” or something like that.
Page 15, Table 2 – Explanation needed – Why 6 CHP is Object A? And why are only 6 CHP while you said before (line 690 page 14) that “27 CHPs will be represented in the sample” ? Explain in article!
Throughout the article there are expressions suitable rather than an oral presentation of the type: “Move on to the experimental part.” , “Now move on to the scale effect.” “These will be the models: CCR and BCC.”, “For example, increase the number of CHPs.” “We can only use it in an output-oriented model. Where this indicator will remain unchanged.” “Now we move on to setting the goal of this study.” “The reason for inefficiency is the inefficient operation of the enterprise in the case of PTE. And the inefficiency is caused by unfavorable conditions in the case of SE.” etc . You must give up this oral expression in a written article and reformulate so that the expression appears continuous for a reader. For example: Instead of : “Move on to the experimental part.” you can say “Next, we will present the experimental part.” or something like that.
Author Response
Please see the file that we attached.

Reviewer 5 Report
Paper presents and interesting study, but some issues such as follows should be addressed:
- in lines 113-114, please revise: "The management concept orientation of any type organization on the effectiveness of its activities requires the"; unfinished sentence;
- in lines 149-152 it is stated that: "For example, it can be imagined that the DMUs under study will be a CHP. At the entrance, it uses the available thermal power of the equipment and the consumption of conventional fuel for the released fuel cell. And as a result of the work it has an output the supply of thermal energy to the grid." ; what about the electricity resulted from CHP?
- in lines 184-185 is specified that "3 main aspects that must be considered when constructing the DEA model"; please revise further the text in order to highlight properly those 3 aspects;
- please pay attention to punctuation (e.g.: line 202, etc.);
- in lines 242-244 "The founders of the model are Charnes, Cooper and Rhodes in 1978 [8]. In this regard, the model was named the CCR model. The name of the model is given as an abbreviation of the authors'."; please revise, it refers to the aforementioned model;
- in line 289, please revise "The presented equation (10) is the BCC-Output Model."; meaning is unclear;
- the quality of Fig. 1 and 2 should be improved; also it is advisable to use English language in fig. 2 text;
- in lines 556 to 562, please revise the text for a better readability;
- in line 567 "The main output indicator of the CHP is the release of thermal energy to the grid"; is it thermal energy or electricity? The association of thermal energy with grid and network terms, especially when CHP units are analyzed is not very accurate. Please explain.
- overall the electricity production of an CHP unit is very important in increasing its yield; apparently, even if the applied method is interesting, it is not clear if the authors understood the basic functionality principles of CHP units.
Minor punctuation and misspelling errors should be revised.
Author Response
Please see the file that we attached.

Round 2
Reviewer 1 Report
All of my concerns have been cleared.
Fine
Reviewer 2 Report
In this revision, the authors have addressed all my concerns and the paper is now acceptable.Reviewer 3 Report
Now the paper can be accepted because author have revised carefully.
please accept this manuscript
Reviewer 4 Report
The authors responded punctually to my observations. I have no other observations.
Reviewer 5 Report
Paper can be published in its current form.